# FEATURE ATTRIBUTION AS FEATURE SELECTION

## ABSTRACT

Feature attribution methods identify "relevant" features as an explanation of a complex machine learning model. Several feature attribution methods have been proposed; however, only a few studies have attempted to define the "relevance" of each feature mathematically. In this study, we formalize the feature attribution problem as a feature selection problem. In our proposed formalization, there arise two possible definitions of relevance. We name the feature attribution problems based on these two relevances as *Exclusive Feature Selection (EFS)* and *Inclusive Feature Selection (IFS)*. We show that several existing feature attribution methods can be interpreted as approximation algorithms for EFS and IFS. Moreover, through exhaustive experiments, we show that IFS is better suited as the formalization for the feature attribution problem than EFS.

## 1 INTRODUCTION

*Feature attribution methods* (Simonyan et al., 2013; Springenberg et al., 2014), or *saliency maps*, are one of the most popular approaches for explaining the decisions of complex machine learning models such as deep neural networks. In feature attribution, for each given instance, the feature attribution methods score how strongly each feature is relevant to the model's decision. An informal definition of the feature attribution problem can be described as follows. We note that this definition is incomplete because the "relevance" of each feature is not defined.

**Feature Attribution Problem** *Given the model $f$ and the target input $x \in \mathbb{R}^d$ to be explained, score $s_i \in \mathbb{R}$ to each feature $x_i$ ($i = 1, 2, \ldots, d$) so that $s_i \geq s_j$ if the feature $x_i$ is more relevant to the model's decision than the feature $x_j$.*

With feature attribution methods, the relevant features can be obtained as explanations why the models made certain decisions. For example, in image recognition, feature attribution methods highlight pixels which the models have focused on, by scoring the relevance of each pixel (Simonyan et al., 2013; Springenberg et al., 2014; Bach et al., 2015; Smilkov et al., 2017), and in text classification, they detect the set of words or sentences relevant to the model's decision by scoring each word or sentence (Ding et al., 2017; Chen et al., 2018).

The major approaches for feature attribution are based on gradient and its modifications (Simonyan et al., 2013; Springenberg et al., 2014; Bach et al., 2015; Smilkov et al., 2017; Shrikumar et al., 2017) and feature occlusions (Zeiler & Fergus, 2014; Zhou et al., 2014).

Most of the studies proposed computational algorithms without defining the "relevance" mathematically (except for some axiomatic approaches (Sundararajan et al., 2017; Lundberg & Lee, 2017)). This means that it is not clear what these algorithm outputs, and we cannot compare these outputs rigorously. To clarify the situation and to establish solid feature attribution methods, we pose the following research questions: (Q1) how can we define relevance? (Q2) is there a general framework for the relevance that induces existing feature attribution methods? and (Q3) what is an appropriate definition of relevance?

In this study, we formalize the feature attribution problem as feature selection problem, and thereby answer questions (Q1)–(Q3). In our proposed formalization, there arise two possible definitions of relevance. We name the feature attribution problems based on these two relevances as *Exclusive Feature Selection (EFS)* and *Inclusive Feature Selection (IFS)*.

Below, we summarize our contributions.

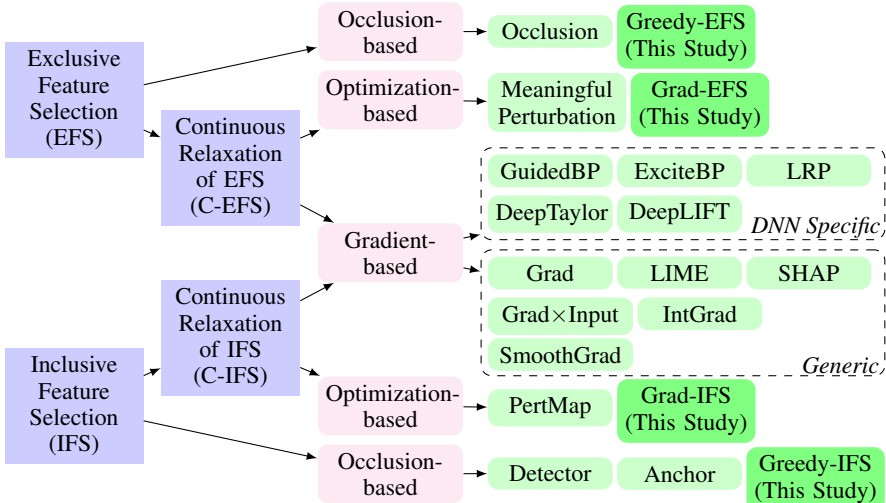

Figure 1: Reorganizing feature attribution methods as Exclusive Feature Selection (EFS) and Inclusive Feature Selection (IFS). See the references for the details of each method: Grad (Simonyan et al., 2013), Grad×Input (Shrikumar et al., 2016), IntGrad (Sundararajan et al., 2017), Smooth-Grad (Smilkov et al., 2017; Hooker et al., 2018), LIME (Ribeiro et al., 2016), SHAP (Lundberg & Lee, 2017), GuidedBP (Springenberg et al., 2014), ExciteBP (Zhang et al., 2016), LRP (Bach et al., 2015), DeepTaylor (Montavon et al., 2017), DeepLIFT (Shrikumar et al., 2017), Occlusion (Zeiler & Fergus, 2014), Detector (Zhou et al., 2014), Anchor (Ribeiro et al., 2018), Meaningful Perturbation (Fong & Vedaldi, 2017), and PertMap (Hara et al., 2018; Ikeno & Hara, 2018)

.

**Answer to Q1: We introduce two formalizations, namely EFS and IFS (Section 2).** We formalize the feature attribution problem as feature selection problem, because the goal of feature attribution is to identify the relevant features to the model's decision. Here, we point out that there are two possible approaches for characterizing the relevance of the features. In the first approach, EFS, we *exclude* some features from the model, and if the model's decision changes by the exclusion, we infer the excluded features are relevant since they have certain impacts to the decision. In the second approach, IFS, we *include* some features to the model, and if the model's decision remains unchanged after the inclusion, we infer the included features are relevant since they are essential to the decision.

**Answer to Q2: The existing methods are based on the relevances of EFS and IFS (Section 3).** We show that several existing feature attribution methods can be interpreted as approximation algorithms for EFS or IFS, as summarized in Figure 1. For example, the gradient-based methods are one-step gradient descent for the continuous relaxation of EFS and IFS.

**Answer to Q3: The relevance based on IFS is better suited for the feature attribution problem (Section 5).** We observe that IFS is better suited as the formalization for the feature attribution problem than EFS. Through exhaustive experiments, we found two crucial properties of EFS. First, the optimal solution to EFS is very similar to adversarial example (Szegedy et al., 2013). As adversarial examples generally provide seemingly meaningless attributions, they are not appropriate for the purpose of explanation. Second, we empirically observe that even a random attribution can perform comparably well with some of the existing feature attribution methods under the evaluation based on EFS. This observation indicates that there are only subtle differences between good attributions and random attributions under EFS. In contrast, unlike EFS, IFS has no trivial drawbacks, and we argue that IFS would be an appropriate formalization for the feature attribution problem.

In this paper, we use the following notation, and consider the problem setting as follows.

**Notation** For any positive integer $d$, $[d]$ denotes the set $[d] = \{1, 2, \ldots, d\}$. We denote $d$-dimensional vectors with all zeros as $0_d$. For a proposition $a$, $\mathbb{I}(a)$ denotes the indicator of $a$, i.e., $\mathbb{I}(a) = 1$ if $a$ is true, and $\mathbb{I}(a) = 0$ if $a$ is false.

**Settings** In this paper, we consider the classification model $f$ for $C$ categories that return an output $y \in \mathbb{R}^C$ for a given data $x \in \mathbb{R}^d$, i.e., $y = f(x)$. The classification result is determined by $c = \operatorname{argmax}_j y_j$ where $y_j = f_j(x)$ is the $j$-th element of the output. We assume that the model $f$ is differentiable with respect to the input $x$: the target models therefore include linear models, kernel models with differentiable kernels, and deep neural networks. We assume that the model $f$ and the target input $x$ to be explained are given and fixed.

## 2 Feature Attribution as Feature Selection

*(Q1) How can we define the relevance?*

As an answer to this question, we formalize the feature attribution problem as feature selection problem, and introduce two definitions of relevance.

Before formalizing the problem, we introduce the idea of *data corruption* (Samek et al., 2017; Fong & Vedaldi, 2017), which plays an important role in this study. Here, we consider corrupting the input data $x$ by overlaying partial features with a noise $r \in \mathbb{R}^d$, as follows.

**Definition 2.1** (Data Corruption). For a vector $x \in \mathbb{R}^d$, the corruption of $x$ with the set $S \subseteq [d]$ and the vector $r \in \mathbb{R}^d$ is given by $x_{S,r}$, which is defined below.

$$(x_{S,r})_i = \begin{cases} r_i & \text{if } i \in S, \\ x_i & \text{otherwise.} \end{cases} \tag{2.1}$$

We refer to the set $S$ as *corrupted features*.

Here, we assume that the noise $r$ follows a distribution $p(r)$. In Section 5, we introduce two types of noises $r$ for the images; we overlay the image to be explained with random noises and random real images.

We now consider the problem formalization. Recall that the objective of feature attribution is to provide high scores to relevant features to the model's decision and low scores to irrelevant features. Our idea is to define the relevance and irrelevance using data corruption. Specifically, in this study, we consider two types of feature selection problems based on data corruption. We name those two problems as *Exclusive Feature Selection (EFS)* and *Inclusive Feature Selection (IFS)*.

### 2.1 Exclusive Feature Selection (EFS)

One way of measuring the relevance of features is to corrupt some of the features by overlaying with uninformative values and observe how the model's decision changes. If the corruption of certain features leads to a decision change, such features can be considered as "relevant". We note that corrupting many features easily leads to a decision change. Therefore, our focus is mainly on a small number of crucial features that can change the decision. We formalize this idea as Exclusive Feature Selection (EFS). In EFS, we aim at changing the decision of the model $f$ to a class different from $c$ by corrupting only a small number of features. See Figure 2 for the idea of EFS. The idea of EFS was originally proposed for measuring the performance of feature attribution methods (Samek et al., 2017). Here, we define EFS as follows.

**Definition 2.2** (Exclusive Feature Selection (EFS)). Find the feature corruption $S \subseteq [d]$ such that (i) the number of corrupted features $|S|$ is small, and (ii) the corrupted data $x_{S,r}$ has small intensity at class $c$, i.e. $f_c(x_{S,r})$ is small, so that the corrupted data is classified into a different class.

$$S_{\text{EFS}} := \operatorname{argmin}_{S \subseteq [d]} |S| + \lambda \mathbb{E}_r[f_c(x_{S,r})], \tag{2.2}$$

where $\mathbb{E}_r$ denotes the expectation over the noise $r$, and $\lambda > 0$ is a weight parameter determined by the user.

In this definition, we consider the expected intensity $\mathbb{E}_r[f_c(x_{S,r})]$ over the noise $r$ so to avoid the corruption to overfit a specific realization of the noise $r$.

By using the solution of EFS, we can define relevance as a binary score as follows. That is, in EFS, the relevant features are the ones when excluded from the data lead to the model's decision change.

**Definition 2.3** (EFS-Relevance). The relevance of each feature $x_i$ is defined by $s_i := \mathbb{I}(i \in S_{\text{EFS}})$.

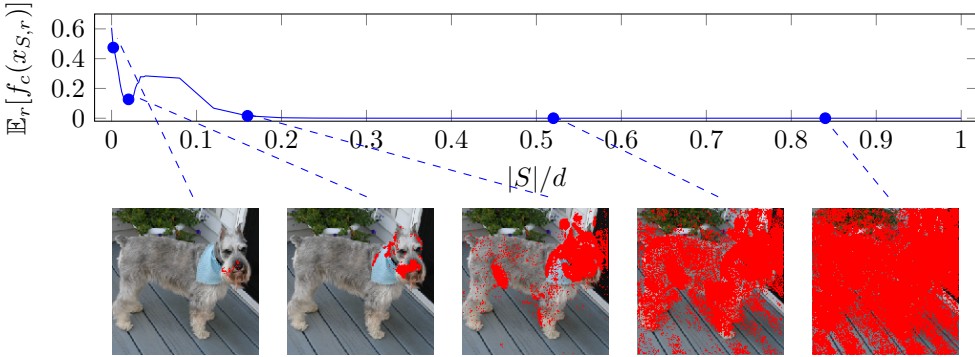

Figure 2: The idea of EFS: Within the trade-off between intensity $\mathbb{E}_r[f_c(x_{S,r})]$ and the number of corrupted features $|S|$, find $S$ that minimizes the intensity and its size. The red pixels indicate the corrupted features $S$. The corrupted features $S$ in the second image is optimal in this curve.

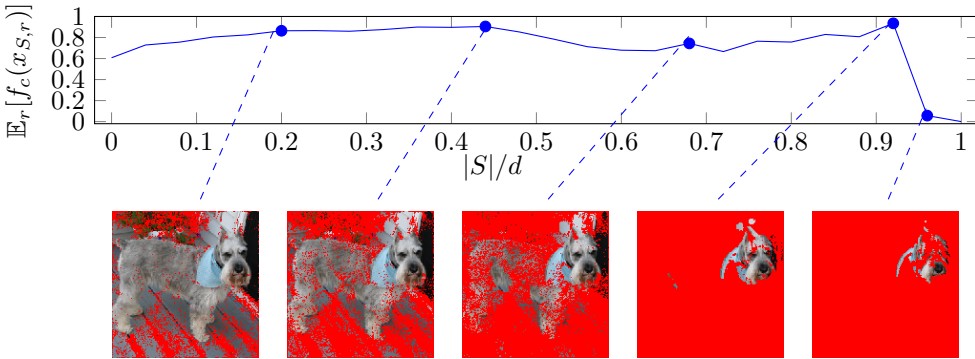

Figure 3: The idea of IFS: Within the trade-off between intensity $\mathbb{E}_r[f_c(x_{S,r})]$ and the number of corrupted features $|S|$, find $S$ that maximizes the intensity and its size. The red pixels indicate the corrupted features $S$. The corrupted features $S$ in the fourth image is optimal in this curve.

## 2.2 INCLUSIVE FEATURE SELECTION (IFS)

Data corruption can be used for measuring the relevance of features in a way different from that of EFS. If the corruption of certain features does not change the model's decision, such features can be considered as "irrelevant". We note that zero corruption trivially keeps the decision unchanged. Therefore, our focus is mainly on a small number of crucial features that have to be kept to maintain the decision. Therefore, in Inclusive Feature Selection, we aim at maintaining the decision of the model $f$ in the class $c$ while corrupting as many features as possible. See Figure 3 for the idea of IFS. Here, we formally define IFS as follows.

**Definition 2.4** (Inclusive Feature Selection (IFS)). Find the feature corruption $S \subseteq [d]$ such that (i) the number of corrupted features $|S|$ is large, and (ii) the corrupted data $x_{S,r}$ has high intensity at class $c$, i.e. $f_c(x_{S,r})$ is large, so that the corrupted data is classified to the class $c$.

$$S_{\text{IFS}} := \operatorname{argmax}_{S \subseteq [d]} |S| + \lambda \mathbb{E}_r[f_c(x_{S,r})]. \tag{2.3}$$

By using the solution of IFS, we can define relevance as follows. In IFS, the relevant features are the ones that when included in the data keep the model's decision unchanged.

**Definition 2.5** (IFS-Relevance). The relevance of each feature $x_i$ is defined by $s_i := \mathbb{I}(i \notin S_{\text{IFS}})$.

## 3 FEATURE ATTRIBUTION METHODS AS EFS AND IFS

*(Q2) Is there a general framework for the relevance that induces existing feature attribution methods?*

As an answer to this question, we show that the existing feature attribution methods can be interpreted as approximation algorithms for EFS or IFS. Thus, the relevances considered in the existing methods are approximated versions of EFS-Relevance and IFS-Relevance. To show this, we classify the existing methods into three types of approaches for solving EFS and IFS: occlusion-based, optimization-based, and gradient-based. See Figure 1 for the overview.

### 3.1 OCCLUSION-BASED APPROACHES

Occlusion-based feature attribution methods (Zeiler & Fergus, 2014; Zhou et al., 2014; Ribeiro et al., 2018) measure the relevance by partially masking features. In those methods, the features are masked by sliding windows or patches, and the change of the output $f_c$ is computed. This can be interpreted as an approximation algorithm for solving the problems (2.2) and (2.3). Instead of searching over exponentially large solution candidates $S \subseteq [d]$, those methods search only over the subset of the solution candidates. For example, one prepares a set of feature subsets $\{S_m : S_m \subseteq [d]\}_{m=1}^M$, and searches for an optimal combination of the subsets by using a greedy algorithm (Zhou et al., 2014) or by a bandit algorithm (Ribeiro et al., 2018).

### 3.2 OPTIMIZATION-BASED APPROACHES

In order to introduce the optimization-based feature attribution methods, we consider the continuous relaxation of EFS and IFS, as follows.

**Definition 3.1** (Continuous Corruption). For a vector $x \in \mathbb{R}^d$, the continuous corruption of $x$ with the vector $w \in [0, 1]^d$ and the vector $r \in \mathbb{R}^d$ is given by $\bar{x}_{w,r}$, which is defined below.

$$(\bar{x}_{w,r})_i = (1 - w_i)x_i + w_i r_i. \tag{3.1}$$

Here, the vector $w$ can be interpreted as the continuous relaxation of the indicator of the set $S$.

**Definition 3.2** (Continuous EFS (C-EFS)). Find the vector $w \in [0, 1]^d$ such that (i) the amount of corruption $\sum_{i=1}^d w_i$ is small, and (ii) the corrupted data $\bar{x}_{w,r}$ has small intensity at class $c$:

$$w_{\text{EFS}} := \text{argmin}_{w \in [0,1]^d} \sum_{i=1}^d w_i + \lambda \mathbb{E}_r[f_c(\bar{x}_{w,r})]. \tag{3.2}$$

**Definition 3.3** (Continuous IFS (C-IFS)). Find the vector $w \in [0, 1]^d$ such that (i) the amount of corruption $\sum_{i=1}^d w_i$ is large, and (ii) the corrupted data $\bar{x}_{w,r}$ has large intensity at class $c$:

$$w_{\text{IFS}} := \text{argmax}_{w \in [0,1]^d} \sum_{i=1}^d w_i + \lambda \mathbb{E}_r[f_c(\bar{x}_{w,r})]. \tag{3.3}$$

For C-EFS and C-IFS, the feature attribution scores can be defined as $s_i = w_{\text{EFS},i}$ and $s_i = 1 - w_{\text{IFS},i}$, respectively. We note that, for a differentiable model $f$, the objective functions of C-EFS (3.2) and C-IFS (3.3) are differentiable. Therefore, these problems can be solved using gradient-based optimization methods such as SGD and Adam (Kingma & Ba, 2014).

Fong & Vedaldi (2017) first introduced the formulation of C-EFS, and they proposed Meaningful Perturbation by adding a smoothness penalty term to C-EFS. PertMap (Hara et al., 2018; Ikeno & Hara, 2018), another optimization-based method, can be interpreted as a variant of C-IFS. PertMap is equivalent to C-IFS with the term $\mathbb{E}_r[f_c(\bar{x}_{w,r})]$ replaced with the hinge penalty term $\sum_{j \neq c} \mathbb{E}_r[\min(0, f_c(\bar{x}_{w,r}) - f_j(\bar{x}_{w,r}))]$ that penalizes $w$ only when the corrupted data is classified into other classes. See Appendix A for the detail.

### 3.3 GRADIENT-BASED APPROACHES

Many feature attribution methods are based on the gradient of the model's output $\frac{\partial f_c(x)}{\partial x_i}$. Here, we point out that those gradient-based feature attribution methods can be interpreted as one-step gradient descent/ascent approximations for C-EFS and C-IFS. If we consider solving the problems

(3.2) and (3.3) using gradient descent/ascent with the initial $w$ set to zeros, i.e. $w = 0_d$, then, the first step of the optimization can be expressed as $w_i \leftarrow 0 \pm \eta \left\{ \mathbb{E}_r \left[ \frac{\partial f_c(\bar{x}_{w,r})}{\partial w_i} \Big|_{w=0_d} \right] + \lambda \right\} \propto$ $\frac{\partial f_c(x)}{\partial x_i}(\mu_i - x_i) + \lambda$, where $\eta > 0$ is the step size and $\mu_i := \mathbb{E}_r[r_i]$. Here, because the penalty $\lambda$ and the step size $\eta$ are common across all the features, $\frac{\partial f_c(x)}{\partial x_i}(\mu_i - x_i)$ is the essential term that determines the size of $w_i$. This finding naturally connects EFS/IFS and the gradient-based feature attribution methods. See Appendix B for the details.

## 4  EFS AND IFS AS EVALUATION METRIC

The ideas of EFS and IFS can be used as metrics for evaluating the performance of feature attribution methods. Suppose that the feature attribution score $s \in \mathbb{R}^d$ is given.

**EFS-based Metric** The principle of EFS is that *"crucial attribution should change the model's decision by corrupting only a small number of relevant features"*. For $q \in [0, 1]$, let the set of the top-$100q\%$ relevant features be $S_q := \{i : s_i \geq t_q\}$, where $t_q$ is the $100q$-th largest percentile in $s$ so that $|S_q| = qd$. Then, we can draw a curve showing the trade-off between the ratio of corrupted relevant features and the degree of the output change (Samek et al., 2017). For example, as the degree of the output changes, we can use the expected difference in intensity: $g_c^{\text{intensity}}(S_q) := \mathbb{E}_r[f_c(x) - f_c(x_{S_q,r})]$, and the probability of label mismatch: $g_c^{\text{label}}(S_q) := \mathbb{E}_r[\mathbb{I}(c \neq \text{argmax}_j f_j(x_{S_q,r}))]$. The trade-off curve usually shows an increasing trend, and the area under the curve can be used as a measure of how quickly the output changes with an increase in the ratio of corrupted features (see Figure 4). In this paper, we refer to this area as *Area Under the EFS-Curve (AUEC)*.

**IFS-based Metric** The principle of IFS is that *"crucial attribution should maintain the model's output even if many irrelevant features are corrupted"*. Similar to EFS, we can construct an IFS-based metric based on this principle. Let $S_q := \{i : s_i \geq t_q\}$ be the top-$100q\%$ relevant features, as defined above. We then corrupt irrelevant features other than $S_q$, i.e. $\bar{S}_q := [d] \setminus S_q$. We can then draw a curve showing the trade-off between the ratio of corrupted irrelevant features and the degree of the output change such as $g_c^{\text{intensity}}(\bar{S}_q)$ and $g_c^{\text{label}}(\bar{S}_q)$. The trade-off curve usually shows an increasing trend with an increase in the ratio of corrupted features (see Figure 4). Therefore, the area over the curve can be used as a measure of how resistant the model's decision is against feature corruption. In this paper, we refer to this area as *Area Over the IFS-Curve (AOIC)*.

## 5  EFS VS. IFS

*(Q3) What is an appropriate definition of relevance?*

To answer this question, we compare EFS and IFS through exhaustive experiments. Our results indicate that EFS has several drawbacks, and we therefore argue that IFS-Relevance is better suited for the feature attribution problem.

### 5.1  EXPERIMENTAL SETUP

**Models and Data** As the target model $f$ to be explained, we adopted three pre-trained models, namely VGG16 (Simonyan & Zisserman, 2014), ResNet V2 with depth 152 (He et al., 2016), and Inception V3 (Szegedy et al., 2016), which were distributed at the Tensorflow repository[1]. As the target data $x$ to be explained, we selected 200 images from the validation set at ILSVRC2014 (Russakovsky et al., 2014) which were correctly classified by the three models.

**Feature Attribution Methods** In the experiments, we adopted several feature attribution methods for comparison: Grad (Simonyan et al., 2013), Grad×Input (Shrikumar et al., 2016), GuidedBP (Springenberg et al., 2014), SmoothGrad (Smilkov et al., 2017; Hooker et al., 2018), IntGrad (Sundararajan et al., 2017), LRP (Bach et al., 2015), DeepLIFT (Shrikumar et al., 2017), Occlusion (Zeiler & Fergus, 2014), and PertMap (Hara et al., 2018; Ikeno & Hara, 2018). Grad,

---

[1]https://github.com/tensorflow/models/tree/master/research/slim

Grad×Input, GuidedBP, SmoothGrad, and IntGrad were implemented using `saliency`[2] with default settings, and LRP, DeepLIFT, and Occlusion were implemented using `DeepExplain`[3], where we set the mask size for Occlusion as $64 \times 64$ with the stride set to 16. We implemented PertMap based on the sample code[4]. We also adopted random attribution as the baseline where the score for each feature was generated uniformly random over $[0, 1]$.

In addition to the existing feature attribution methods, we implemented the following EFS-based and IFS-based methods: *Greedy-EFS*, which solves the problem (2.2) using a greedy algorithm; *Grad-EFS*, which solves the problem (3.2) using gradient descent; *Greedy-IFS*, which solves the problem (2.3) using a greedy algorithm; and *Grad-IFS*, which solves the problem (3.3) using gradient ascent. The details of these methods can be found in Appendix C.

**Evaluation** For evaluating AUEC and AOIC, we prepared two noise distributions $p(r)$. The first distribution is a uniform distribution: each $r$ is independently sampled from the uniform distribution over $[0, 1]^d$. The second distribution is a distribution over real images. We selected 100 images from the validation set at ILSVRC2014, with no overlap with the 200 images to be explained. Then, from those 100 images, we randomly selected an image as the noise $r$. To compute AUEC and AOIC, we varied the percentile $q$ from zero to one, and for each $q$, we computed the difference scores $g_c^{\text{label}}(S_q)$ and $g_c^{\text{label}}(\bar{S}_q)$ using empirical averages under those two noise distributions.

## 5.2 RESULTS

For each model, we computed the attribution scores for all 200 images using each of the 14 feature attribution methods. We then computed AUEC and AOIC under the two noise distributions.

Our main result is summarized in Table 1. Table 1 shows the AUEC and AOIC for VGG16 for the 200 images under uniform noise. We moved the results for the other models and the corruption with real images to Appendix D, as those results are similar. Here, we point out that there are three important observations in the table.

**EFS found adversarial example.** Grad-EFS attained the highest AUEC. This indicates that Grad-EFS is nearly optimal under the principle of EFS: Grad-EFS can change the model's decision by corrupting only a small number of relevant features. Indeed, as shown in Figure 4, Grad-EFS has a sharp increase in the EFS-Curve [5]. Specifically, it shows that Grad-EFS successfully changed the model's decision for more than 80% of the data by corrupting only a few percent of the pixels. Similar tendencies were also observed for ResNet V2 and Inception V3 (see Appendix D).

An important observation is that the heatmap of Grad-EFS is just a shot noise, as shown in Figure 5. This is because EFS is very similar to adversarial example (Szegedy et al., 2013). In adversarial example, one seeks the minimum data perturbation that changes the model's output. In EFS (2.2), instead of the data perturbation, one searches for a small number of corrupted features that reduces the class intensity. Similarly, C-EFS (3.2) searches for a continuous corruption with the minimum $\ell_1$ norm.

**Random attribution performed comparably well with existing methods.** The random attribution attained AUEC similar to that of methods such as Grad×Input, IntGrad, LRP, and Occlusion, especially for VGG16. Indeed, as shown in Figure 4, the EFS-Curve of random attribution is close to those methods. It is a bit surprising to observe that we can attain a good trade-off in EFS just by randomly scoring each feature without looking at the images. This observation indicates that there are only subtle differences between good attributions and random attributions under EFS, especially for VGG16.

**Grad-IFS significantly outperformed the other methods.** On AOIC, Grad-IFS significantly outperformed the other methods, and PertMap attained the second best result. As shown in Figure 4, Grad-IFS is very resistant against the corruption of irrelevant features. Indeed, even if 80% of the pixels are corrupted, the model's decision is kept unchanged for more than 80% of the images. This means that Grad-IFS is capable of identifying irrelevant features better than any other methods. An-

---

[2]`https://github.com/PAIR-code/saliency`

[3]`https://github.com/marcoancona/DeepExplain`

[4]`https://github.com/sato9hara/PertMap/`

[5]We selected nine feature attribution methods for the visibility purpose.

Table 1: Average AUEC and AOIC under the uniform noise. The top-three scores are highlighted as 1st*, 2nd**, and 3rd***.

| | VGG16 | |
| --- | --- | --- |
| | AUEC | AOIC |
| Greedy-EFS | 0.844 | 0.366 |
| Grad-EFS | 0.946* | 0.195 |
| Greedy-IFS | 0.746 | 0.622*** |
| Grad-IFS | 0.873 | 0.876* |
| Grad | 0.867 | 0.341 |
| Grad×Input | 0.823 | 0.318 |
| SmoothGrad | 0.882 | 0.593 |
| GuidedBP | 0.918** | 0.455 |
| IntGrad | 0.837 | 0.346 |
| LRP | 0.823 | 0.318 |
| DeepLIFT | 0.862 | 0.435 |
| Occlusion | 0.811 | 0.559 |
| PertMap | 0.886*** | 0.780** |
| Random | 0.839 | 0.160 |

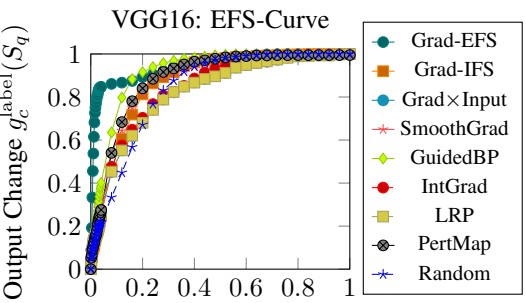

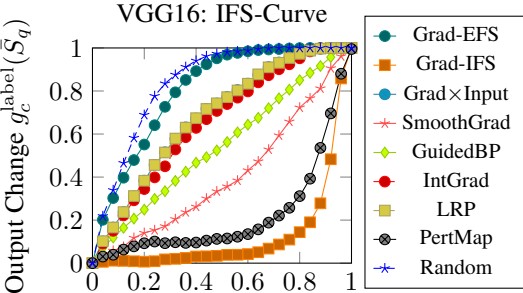

Figure 4: Average EFS-Curve and IFS-Curve.

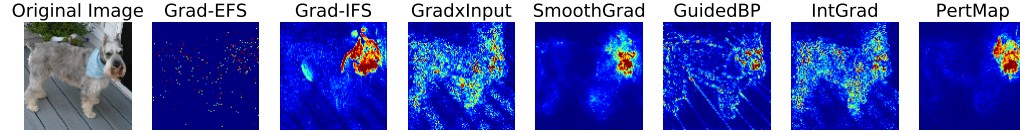

Figure 5: Attributions on VGG16: The red colored pixels are found to be strongly relevant with each method.

other interesting point that can be seen in Figure 4 is that, the IFS-Curves vary significantly across different methods. More importantly, unlike EFS, the IFS-Curve can distinguish random attributions and other attributions well.

Figure 5 shows the examples of the attributions obtained by each method. It is important to note that the top-three AOIC methods, namely Grad-IFS, PertMap, and SmoothGrad, have highlighted only the dog face. The high AOICs on these methods indicate that the model has made the decision based on the dog face. In contrast, the other methods tend to generate noisy attributions over the entire body of the dog, which are false explanations from the perspective of IFS because their AOICs are far smaller than that of Grad-IFS: the noisy attributions failed to capture essential pixels in the image.

## 6 CONCLUSION

In this study, we formalized the feature attribution problem as two types of feature selection problems, which we named as EFS and IFS. Based on EFS and IFS, we clarified that the existing feature attribution methods can be interpreted as approximation algorithms for EFS and IFS. Then, through exhaustive experiments, we clarified that IFS is better suited as the formalization for the feature attribution problem; we observed that EFS has several unfavorable properties and concluded that EFS is not an appropriate formalization.

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

# SUPPLEMENTARY MATERIAL FOR "FEATURE ATTRIBUTION AS FEATURE SELECTION"

**Preliminaries**    We define the data-dependent noise by $r_x$ as follows:

$$r_x := x + u,$$

where we assume that the perturbation $u$ follows a distribution $p(u)$ with $\mathbb{E}_u[u] = 0$. The continuous corruption with the noise $r_x$ is then expressed as

$$(\bar{x}_{w,r_x})_i = (1 - w_i)x_i + w_i r_{x,i} = x_i + w_i u_i.$$

That is, the continuous corruption $\bar{x}_{w,r_x}$ is centered at the data $x$. Moreover, $\bar{x}_{w,r_x}$ distributes around $x$ with the magnitude $w$. The data-dependent noise and this continuous corruption plays an important role when interpreting some of the existing feature attribution methods as EFS and IFS.

## A    OPTIMIZATION-BASED FEATURE ATTRIBUTION METHODS AS EFS/IFS

Meaningful Perturbation (Fong & Vedaldi, 2017) is a variant of C-EFS defined below:

$$\min_{w \in [0,1]^d} \sum_{i=1}^d w_i + \lambda f_c(\bar{x}_{w,r}) + \phi_{\text{smooth}}(w). \tag{A.1}$$

The differences from C-EFS (3.2) are (i) it uses a fixed noise $r$ instead of the expectation, and (ii) the additional penalty term $\phi_{\text{smooth}}(w)$ is added so that $w$ to be smooth. Here, the smoothness penalty is essential for Meaningful Perturbation. Fong & Vedaldi (2017) have reported that the solution to the problem (A.1) without the smoothness penalty tends to be seemingly meaningless attributions. This is because C-EFS is very similar to the $\ell_1$ penalized adversarial example, as we have discussed in Section 5. Therefore, we need to design an appropriate penalty term $\phi_{\text{smooth}}$ to obtain reasonable attributions using Meaningful Perturbation.

PertMap (Hara et al., 2018; Ikeno & Hara, 2018) can be interpreted as a variant of C-IFS. PertMap finds irrelevant features by maximizing the data perturbation. Let $\bar{x}_{w,r_x}$ be the continuous corruption of the data $x$ with a data-dependent noise $r_x$, where the parameter $w$ determine the magnitude of the data perturbation. In PertMap, one seeks for the maximum data perturbation that maintains the classification result unchanged from the original class $c$. The optimization problem of $w$ is defined as follows (Ikeno & Hara, 2018):

$$\max_{w \in [0,1]^d} \sum_{i=1}^d w_i + \lambda \sum_{j \neq c} \mathbb{E}_r[\min(0, f_c(\bar{x}_{w,r}) - f_j(\bar{x}_{w,r}))]. \tag{A.2}$$

This problem is equivalent to C-EFS except for the term $\mathbb{E}_r[f_c(\bar{x}_{w,r})]$ replaced with the hinge penalty term $\sum_{j \neq c} \mathbb{E}_r[\min(0, f_c(\bar{x}_{w,r}) - f_j(\bar{x}_{w,r}))]$ that penalizes $w$ only when the corrupted data is classified into other classes.

## B    GRADIENT-BASED FEATURE ATTRIBUTION METHODS AS EFS/IFS

Many feature attribution methods are based on the gradient of the model's output $\frac{\partial f_c(x)}{\partial x_i}$. Here, we show that those gradient-based feature attribution methods can be interpreted as one-step gradient descent/ascent approximations to C-EFS and C-IFS. If we consider solving the problems (3.2) and (3.3) using gradient descent/ascent with the initial $w$ set to zeros, i.e. $w = 0_d$, then, the first step of the optimization can be expressed as

$$w_i \leftarrow 0 \pm \eta \left\{ \mathbb{E}_r \left[ \left. \frac{\partial f_c(\bar{x}_{w,r})}{\partial w_i} \right|_{w=0_d} \right] + \lambda \right\} \propto \frac{\partial f_c(x)}{\partial x_i}(\mu_i - x_i) + \lambda, \tag{B.1}$$

where $\eta > 0$ is the step size and $\mu_i := \mathbb{E}_r[r_i]$. Here, because the penalty $\lambda$ and the step size $\eta$ are common across all the features, $\frac{\partial f_c(x)}{\partial x_i}(\mu_i - x_i)$ is the essential term that determines the size of $w_i$. We therefore ignore the terms $\lambda$ and $\eta$ for simplicity. Then, the relationship between EFS/IFS and the gradient-based feature attribution methods can be summarized as follows.

**Grad (Simonyan et al., 2013)** Assuming $\mu_i - x_i$ to be constant across the features, $w_i$ is equivalent to $\frac{\partial f_c(x)}{\partial x_i}$ up to the scaling factor $\mu_i - x_i$.

**Grad×Input (Shrikumar et al., 2016)** Assuming $\mu_i$ to be sufficiently small ($\mu_i \rightarrow 0$), $w_i$ is equivalent to $\frac{\partial f_c(x)}{\partial x_i} x_i$.

**Linear Approximation (Bach et al., 2015; Sundararajan et al., 2017; Shrikumar et al., 2017; Montavon et al., 2017; Ribeiro et al., 2016; Lundberg & Lee, 2017)** Several methods consider the linear approximation of the model $f$ in the neighborhood of the input $x$. Specifically, they consider the linear model

$$f_c(r) = f_c(x) + \langle w, r - x \rangle, \tag{B.2}$$

and uses the coefficient $w$ as the attribution score. Here, let us assume that the noise $r$ belongs to the $\epsilon$-ball around $x$, i.e. $r \in R(x; \epsilon) := \{r : \|r - x\| \leq \epsilon\}$. For differentiable models, we can consider the first-order Taylor expansion within $R(x; \epsilon)$:

$$f_c(r) = f_c(x) + \langle \nabla f_c(x), r - x \rangle + O(\epsilon^2). \tag{B.3}$$

Thus, in a rough sense, by ignoring the $O(\epsilon^2)$ term, the coefficient $w_i$ computed by those linear approximation methods is essentially equivalent to the gradient $\frac{\partial f_c(x)}{\partial x_i}$. Technically, those linear approximation methods consider further improvements over the pure gradient, so that the linear approximation to be valid not only in the infinitesimally small neighborhood of $x$, but in a finite range from $x$. Thus, the computed coefficient $w_i$ can slightly differ from the gradient, nevertheless, those methods can be still classified as approximations of the gradient.

**SmoothGrad (Smilkov et al., 2017; Hooker et al., 2018)** In SmoothGrad, the attribution score is defined as the expectation of the squared gradient over perturbed inputs. Recall that the perturbed data can be expressed as $\bar{x}_{w, r_x}$ using the data-dependent noise $r_x$. Given the perturbation magnitude $w_0 > 0$, the attribution score is defined as follows (Smilkov et al., 2017; Hooker et al., 2018):

$$\text{score}_{\text{SG}} := \mathbb{E}_r \left[ \left( \frac{\partial f_c(\bar{x}_{w_0, r_x})}{\partial (\bar{x}_{w_0, r_x})_i} \right)^2 \right] \tag{B.4}$$

Here, we show that this score can be also interpreted as one-step gradient descent/ascent approximation to C-EFS and C-IFS. If we set the initial $w$ as $w_i = w_0$, the first step of the optimization can be expressed as

$$w_i \leftarrow w_0 \pm \eta \left\{ \mathbb{E}_r \left[ \left. \frac{\partial f_c(\bar{x}_{w, r_x})}{\partial w_i} \right|_{w=w_0} \right] + \lambda \right\} \tag{B.5}$$

$$= \pm \eta \mathbb{E}_r \left[ \left. \frac{\partial f_c(\bar{x}_{w, r_x})}{\partial w_i} \right|_{w=w_0} \right] + (w_0 \pm \eta \lambda) \tag{B.6}$$

$$= \pm \eta \mathbb{E}_u \left[ \left. \frac{\partial f_c(\bar{x}_{w, x+u})}{\partial (\bar{x}_{w, x+u})_i} \right|_{w=w_0} u_i \right] + (w_0 \pm \eta \lambda), \tag{B.7}$$

where we used the definition $r_x = x + u$ in the last equality. Because $w_0$, $\eta$, and $\lambda$ are constants, the first term determines $w_i$. Moreover, by applying the Cauchy-Schwartz inequality, we obtain

$$\left( \mathbb{E}_u \left[ \left. \frac{\partial f_c(\bar{x}_{w_0, x+u})}{\partial (\bar{x}_{w_0, x+u})_i} \right|_{w=w_0} u_i \right] \right)^2 \leq \mathbb{E}_r \left[ \left( \frac{\partial f_c(\bar{x}_{w_0, r_x})}{\partial (\bar{x}_{w_0, r_x})_i} \right)^2 \right] \mathbb{E}_u[u_i^2]. \tag{B.8}$$

If we selected the perturbation $u$ to have a common variance across the features, i.e. $\mathbb{E}_u[u_i^2] = \sigma^2$ for all $i \in [d]$, we can conclude that $\mathbb{E}_r \left[ \left( \frac{\partial f_c(\bar{x}_{w_0, r_x})}{\partial (\bar{x}_{w_0, r_x})_i} \right)^2 \right]$ is the essential term determining $w$, which is equivalent to (B.4).

Here, we note out our analysis can explain the success of SmoothGrad. Some studies (Smilkov et al., 2017; Hooker et al., 2018) have reported that SmoothGrad performs better than other gradient-based

feature attribution methods in practice. As one-step gradient descent/ascent approximation to C-EFS and C-IFS, SmoothGrad starts the optimization from the non-zero point $w = w_0$ while the other methods starts the optimization from zero $w = 0_d$. If the initial point $w_0$ is carefully chosen, it is apparent that one-step gradient descent/ascent from $w_0$ can get closer to the solution than starting the optimization from zero.

## C   Implementations of EFS/IFS-based Methods

In Section 5, we adopted the EFS-based and IFS-based methods: *Greedy-EFS*, which solves the problem (2.2) using a greedy algorithm; *Grad-EFS*, which solves the problem (3.2) using gradient descent; *Greedy-IFS*, which solves the problem (2.3) using a greedy algorithm; and *Grad-IFS*, which solves the problem (3.3) using gradient ascent. Each method is implemented as follows.

**Greedy-EFS**   In Greedy-EFS, we first prepared the feature subsets $\mathcal{S} := \{S_m \subseteq [d]\}_{m=1}^M$, where each subset $S_m$ is constructed by sliding the window over the image. In our implementation, we set the window size as $64 \times 64$ and the stride set to $16$. We started the greedy algorithm from $S = \emptyset$, and iterated the following steps.

1. $\hat{S}_m := \operatorname{argmin}_{S_m \in \mathcal{S}} \mathbb{E}_r[f_c(x_{S \cup S_m, r})]$ .
2. $S \leftarrow S \cup \hat{S}_m, \mathcal{S} \leftarrow \mathcal{S} \setminus \{\hat{S}_m\}$.

We repeated those steps ten times, and obtained $S$ as an approximation of $S_{\text{EFS}}$. We used a gray background as the noise $r$, i.e. $p(r) = \delta(r = \text{gray})$.

**Greedy-IFS**   In Greedy-IFS, we prepared the feature subsets $\mathcal{S} := \{S_m \subseteq [d]\}_{m=1}^M$ in the same way as Greedy-EFS. We started the greedy algorithm from $S = [d]$, and iterated the following steps.

1. $\hat{S}_m := \operatorname{argmax}_{S_m \in \mathcal{S}} \mathbb{E}_r[f_c(x_{S \setminus S_m, r})]$ .
2. $S \leftarrow S \setminus \hat{S}_m, \mathcal{S} \leftarrow \mathcal{S} \setminus \{\hat{S}_m\}$.

We repeated those steps ten times, and obtained $S$ as an approximation of $S_{\text{IFS}}$. We used a gray background as the noise $r$, i.e. $p(r) = \delta(r = \text{gray})$. We note that Greedy-IFS is essentially the same as the greedy method proposed by Zhou et al. (2014).

**Grad-EFS**   In Grad-EFS, we used the data-dependent noise $r_x$, where we set the distribution $p(u)$ to be uniform over $[-1, 1]^d$. In each step of the gradient descent, we approximated the gradient of C-EFS (3.2) using a batch of random realizations of $r_x$. In our implementation, we set the penalty weight $\lambda = 10d$ and the batch size in each gradient approximation to be $32$. As the optimization algorithm, we used Adam (Kingma & Ba, 2014) with the step size set to $0.5$ and the remaining parameters set to default values. We run Adam for 200 steps, and obtained $w$. We note that Grad-EFS is essentially the same as Meaningful Perturbation (Fong & Vedaldi, 2017) except that the smoothness penalty term is removed.

**Grad-IFS**   In Grad-IFS, we used the data-dependent noise $r_x$, where we set the distribution $p(u)$ to be uniform over $[-1, 1]^d$. In each step of the gradient ascent, we approximated the gradient of C-IFS (3.3) using a batch of random realizations of $r_x$. In our implementation, we set the penalty weight $\lambda = d$ and the batch size in each gradient approximation to be $32$. As the optimization algorithm, we used Adam (Kingma & Ba, 2014) with the step size set to $0.03$ and the remaining parameters set to default values. We run Adam for 1000 steps. In Grad-IFS, we used the averaged parameter over 1000 steps $\frac{1}{1000} \sum_{t=1}^{1000} w^{(t)}$ as $w$. This is a common technique to improve the quality of the solution obtained from stochastic optimization algorithms with convex objective functions. We found that this technique is helpful also for Grad-IFS. We also applied the same averaging technique to PertMap (Ikeno & Hara, 2018).

## D   Experimental Results

Here, we present all the results that are omitted from Section 5 due to the space limitation.

### D.1 AUEC AND AOIC

Table 2 and Table 3 show average AUEC and AOIC evaluated with uniform noises and random images, respectively. There are three important observations in these tables.

First, Grad-EFS attained the highest AUEC for VGG16 and Inception V3, and the second highest AUEC for ResNet V2. This indicates that Grad-EFS is nearly optimal under the principle of EFS: Grad-EFS can change the model's decision by corrupting only a small number of relevant features. Indeed, as shown in Figure 6 and Figure 7, Grad-EFS has a sharp increase in the EFS-Curves. This is a natural consequence from the fact that EFS is very similar to adversarial example, as we discussed in Section 5.

Second, especially for VGG16, the random attribution attained AUEC similar to some of the methods such as Grad×Input, IntGrad, LRP, and Occlusion. Indeed, as in Figure 6 and Figure 7, the EFS-Curves of the random attribution are close to those methods. It is a bit surprising to observe that we can attain a good trade-off in EFS just by randomly scoring each feature without looking at the images. This observation indicates that there are only subtle differences between good attributions and random attributions under EFS, especially for VGG16.

Third, on AOIC, Grad-IFS consistently outperformed the other methods, and PertMap attained the second best result. As shown in Figure 6 and Figure 7, Grad-IFS is very resistant against the corruption of irrelevant features. This means that Grad-IFS is capable of identifying irrelevant features better than any other methods. Another important point that can be seen in Figure 6 and Figure 7 is that, the IFS-Curves vary significantly across different methods. More importantly, unlike EFS, the IFS-Curve can distinguish random attributions and other attributions well.

Table 2: Average AUEC and AOIC under the uniform noise. The top-three scores are highlighted as 1st*, 2nd**, and 3rd***.

| | VGG16 | | ResNet V2 | | Inception V3 | |
|---|---|---|---|---|---|---|
| | AUEC | AOIC | AUEC | AOIC | AUEC | AOIC |
| Greedy-EFS | 0.844 | 0.366 | 0.858** | 0.391 | 0.837*** | 0.445 |
| Grad-EFS | 0.946* | 0.195 | 0.858** | 0.369 | 0.927* | 0.394 |
| Greedy-IFS | 0.746 | 0.622 | 0.745 | 0.709*** | 0.715 | 0.685 |
| Grad-IFS | 0.873 | 0.876* | 0.814 | 0.829* | 0.817 | 0.868* |
| Grad | 0.867 | 0.341 | 0.797 | 0.416 | 0.799 | 0.475 |
| Grad×Input | 0.823 | 0.318 | 0.750 | 0.404 | 0.748 | 0.441 |
| SmoothGrad | 0.882 | 0.593*** | 0.840 | 0.707 | 0.829 | 0.744*** |
| GuidedBP | 0.918** | 0.455 | 0.874* | 0.595 | 0.826 | 0.555 |
| IntGrad | 0.837 | 0.346 | 0.781 | 0.438 | 0.775 | 0.482 |
| LRP | 0.823 | 0.318 | 0.750 | 0.404 | 0.748 | 0.441 |
| DeepLIFT | 0.862 | 0.435 | 0.811 | 0.518 | 0.788 | 0.539 |
| Occlusion | 0.811 | 0.559 | 0.767 | 0.595 | 0.709 | 0.664 |
| PertMap | 0.886*** | 0.780** | 0.828 | 0.765** | 0.846** | 0.817** |
| Random | 0.839 | 0.160 | 0.691 | 0.308 | 0.714 | 0.290 |

Table 3: Average AUEC and AOIC under random images. The top-three scores are highlighted as 1st*, 2nd**, and 3rd***.

| | VGG16 | | ResNet V2 | | Inception V3 | |
|---|---|---|---|---|---|---|
| | AUEC | AOIC | AUEC | AOIC | AUEC | AOIC |
| Greedy-EFS | 0.842 | 0.319 | 0.857*** | 0.388 | 0.857*** | 0.397 |
| Grad-EFS | 0.926* | 0.179 | 0.861** | 0.275 | 0.893* | 0.293 |
| Greedy-IFS | 0.766 | 0.524*** | 0.763 | 0.589 | 0.760 | 0.521 |
| Grad-IFS | 0.858 | 0.630* | 0.826 | 0.688* | 0.844 | 0.705* |
| Grad | 0.865 | 0.290 | 0.800 | 0.375 | 0.826 | 0.393 |
| Grad×Input | 0.835 | 0.296 | 0.775 | 0.384 | 0.789 | 0.396 |
| SmoothGrad | 0.880 | 0.486 | 0.846 | 0.647*** | 0.860 | 0.650*** |
| GuidedBP | 0.905** | 0.416 | 0.876* | 0.568 | 0.839 | 0.502 |
| IntGrad | 0.847 | 0.318 | 0.807 | 0.418 | 0.815 | 0.435 |
| LRP | 0.835 | 0.296 | 0.775 | 0.384 | 0.789 | 0.397 |
| DeepLIFT | 0.868 | 0.378 | 0.836 | 0.480 | 0.826 | 0.471 |
| Occlusion | 0.840 | 0.487 | 0.787 | 0.544 | 0.769 | 0.553 |
| PertMap | 0.876*** | 0.553** | 0.833 | 0.660** | 0.861** | 0.661** |
| Random | 0.842 | 0.160 | 0.760 | 0.237 | 0.767 | 0.234 |

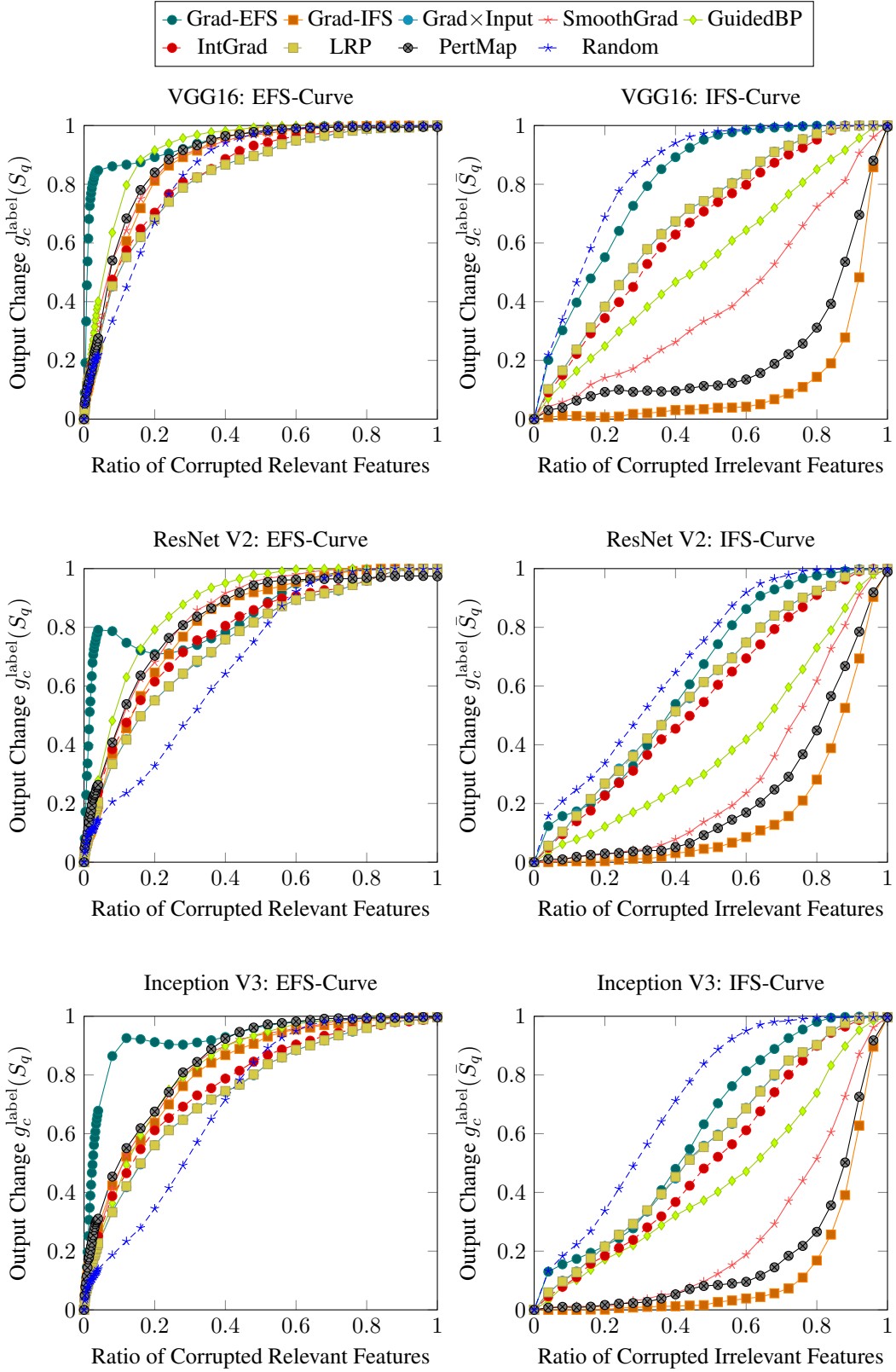

Figure 6: Average EFS-Curve and IFS-Curve evaluated with uniform noises.

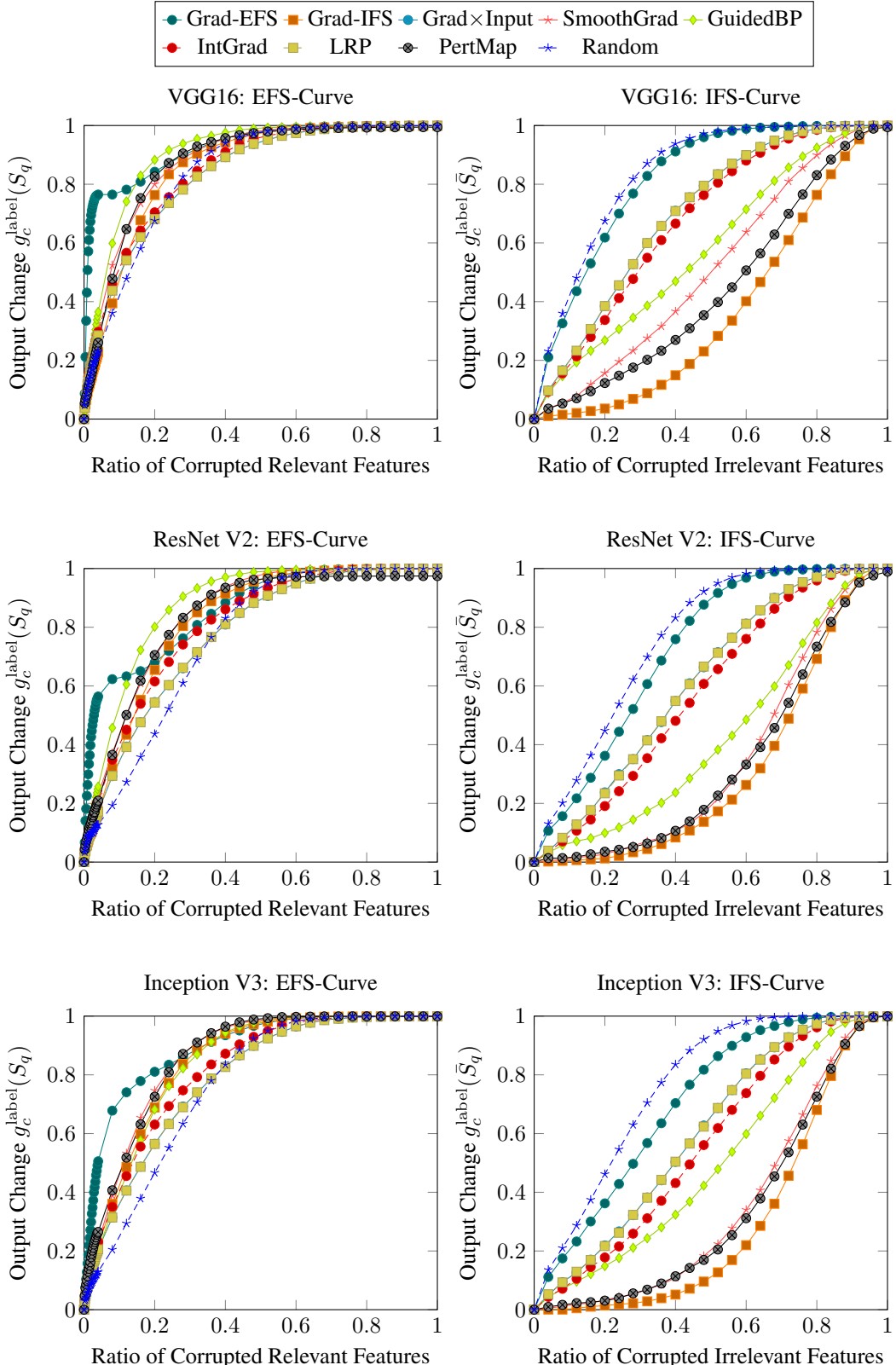

Figure 7: Average EFS-Curve and IFS-Curve evaluated with random images.

## D.2   EXAMPLE HEATMAPS

Figures 8–12 show examples of the heatmaps obtained by several feature attribution methods. There are two important observations in the figures.

First, the heatmaps of Grad-EFS are mostly shot noises, which are visibly meaningless. These are natural consequences from the fact that EFS is very similar to adversarial example, as we have discussed in Section 5.

Second, the high AOIC methods in Table 2 and Table 3, namely Grad-IFS, PertMap, and Smooth-Grad, have highlighted mostly the faces of the animals. The high AOICs on these methods indicate that the model has made the decision based mostly on the animal face. In contrast, the other methods tend to generate noisy attributions over the entire body of the animals, which are false explanations from the perspective of IFS because their AOICs are far smaller than that of Grad-IFS: the noisy attributions failed to capture essential pixels in the images.

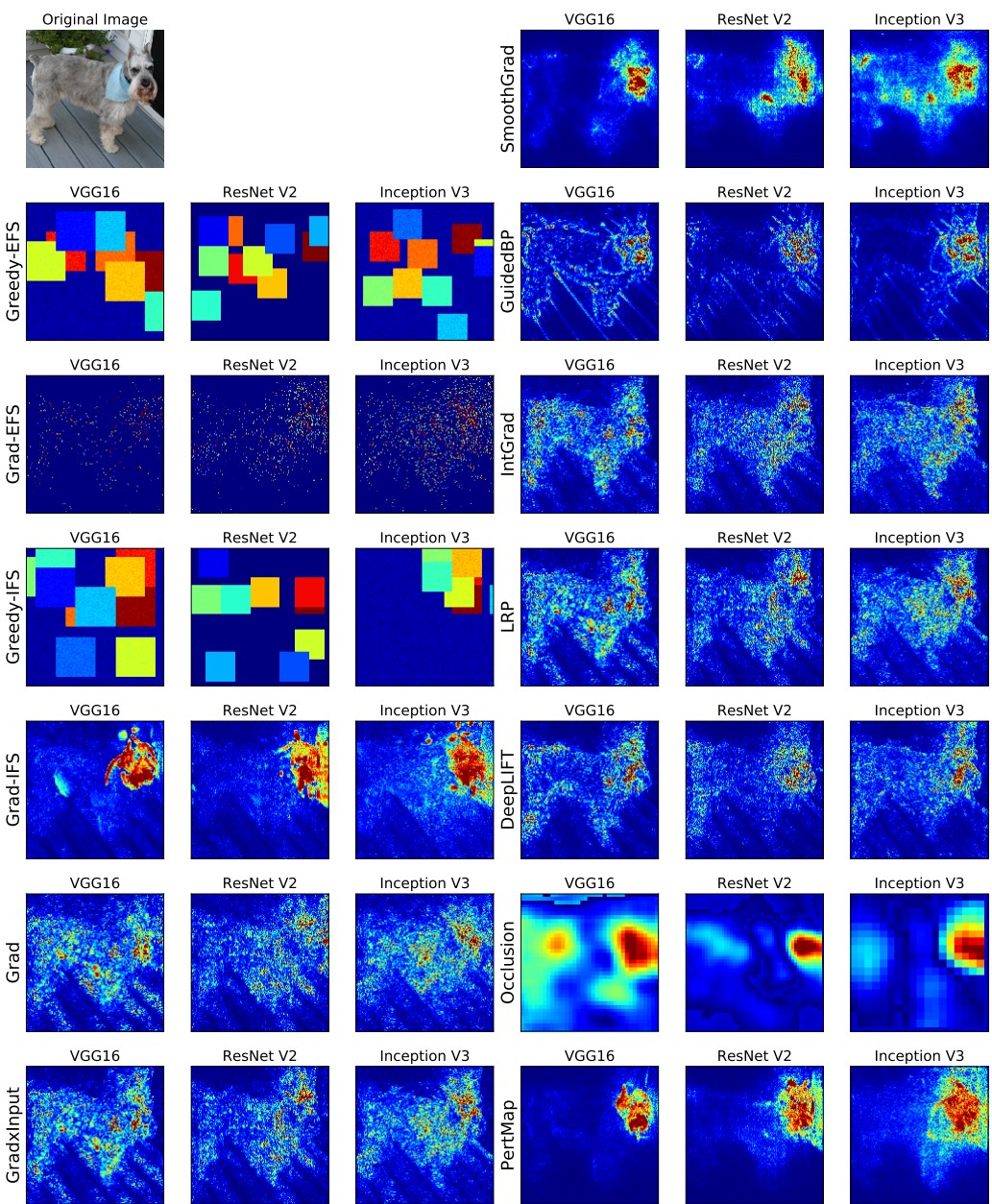

Figure 8: Example Heatmaps: The red colored pixels are found to be strongly relevant with each method.

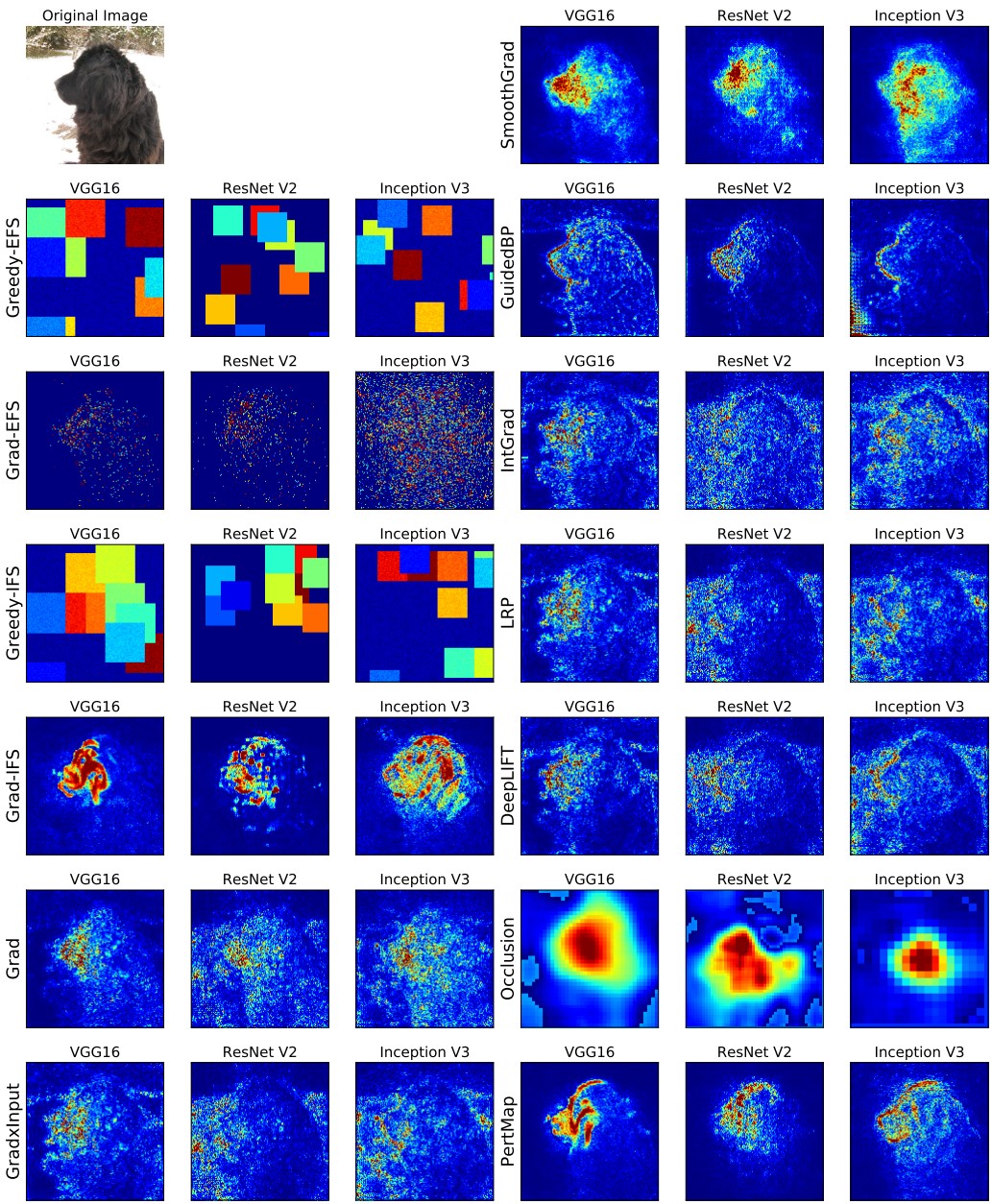

Figure 9: Example Heatmaps: The red colored pixels are found to be strongly relevant with each method.

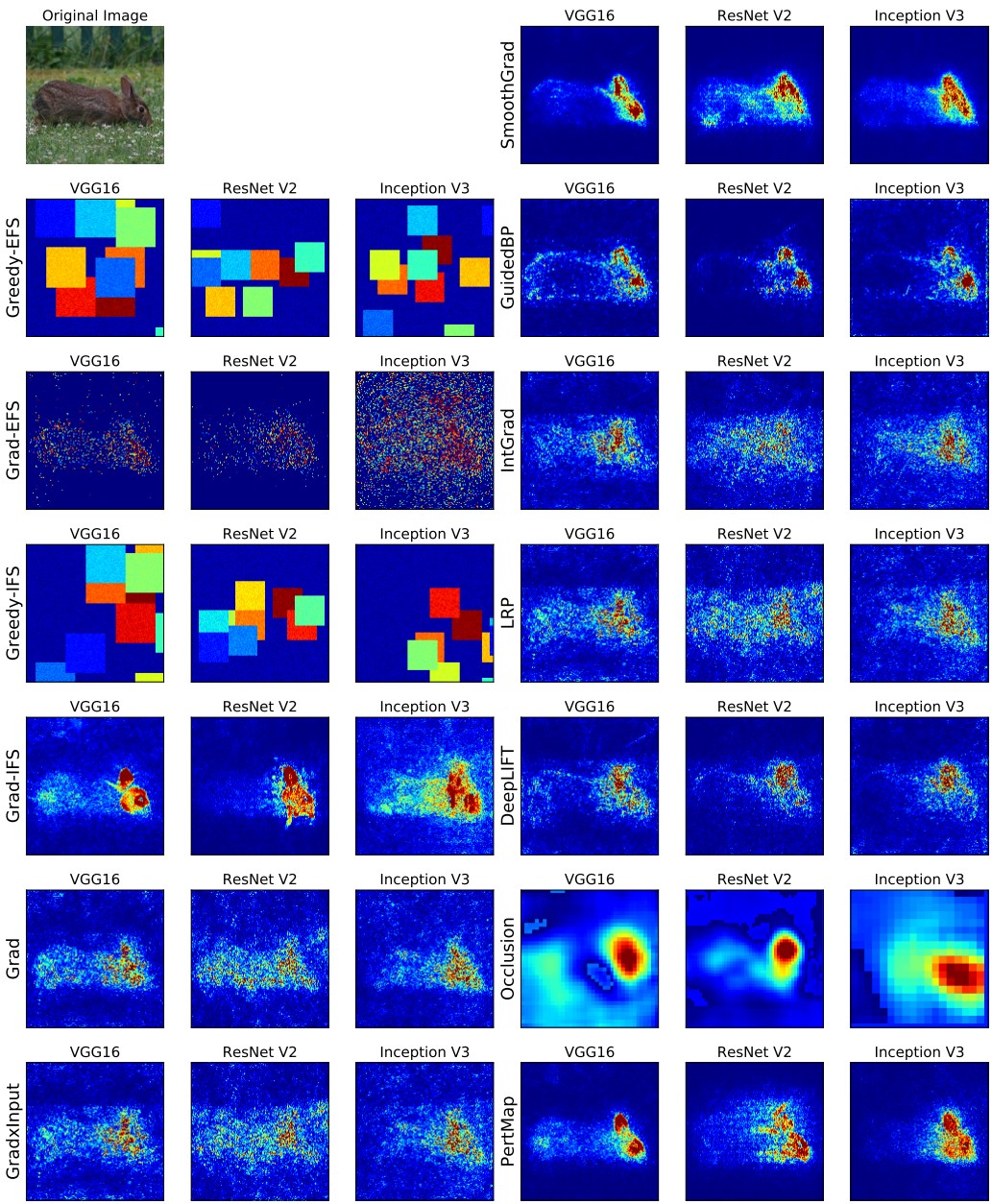

Figure 10: Example Heatmaps: The red colored pixels are found to be strongly relevant with each method.

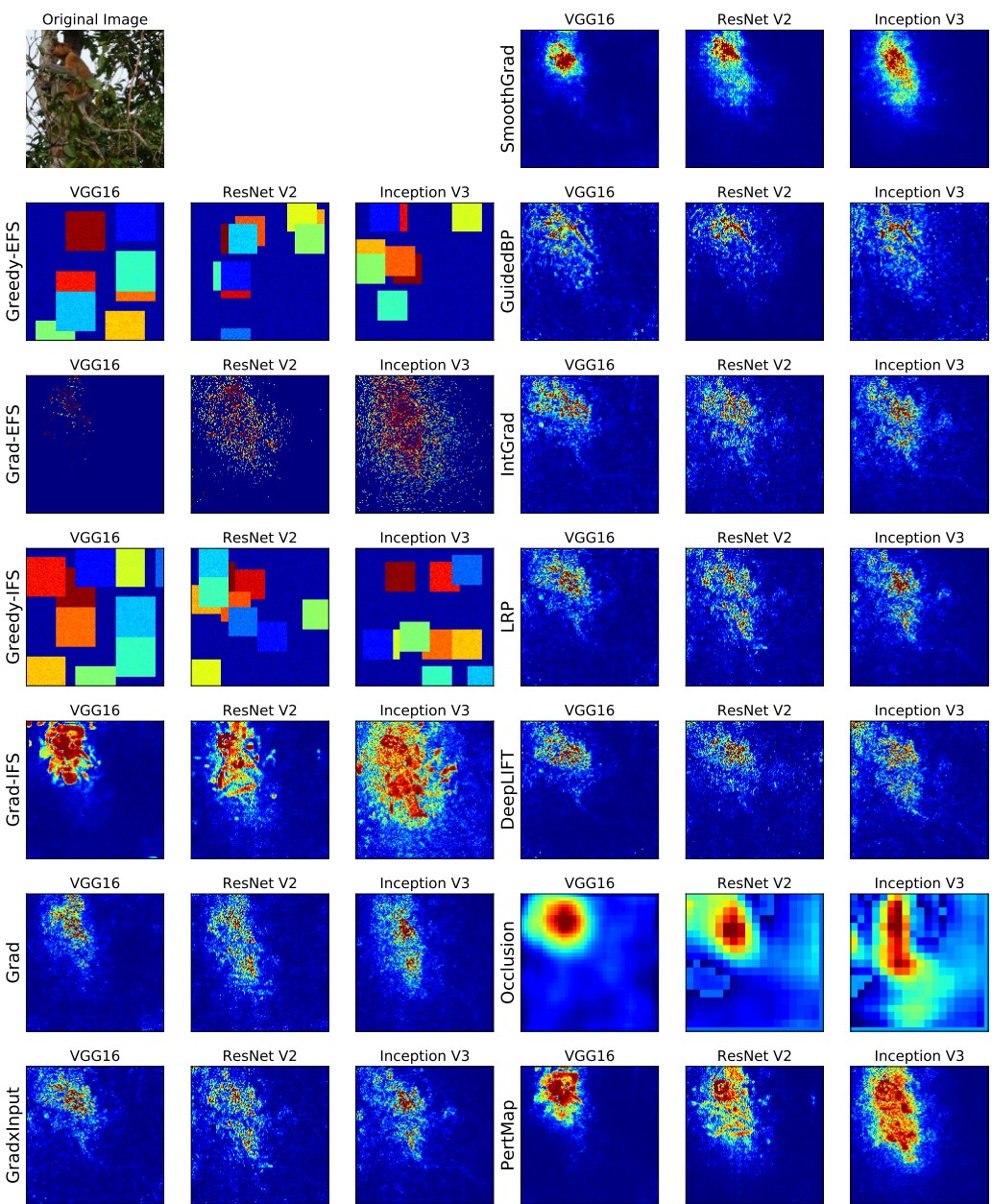

Figure 11: Example Heatmaps: The red colored pixels are found to be strongly relevant with each method.

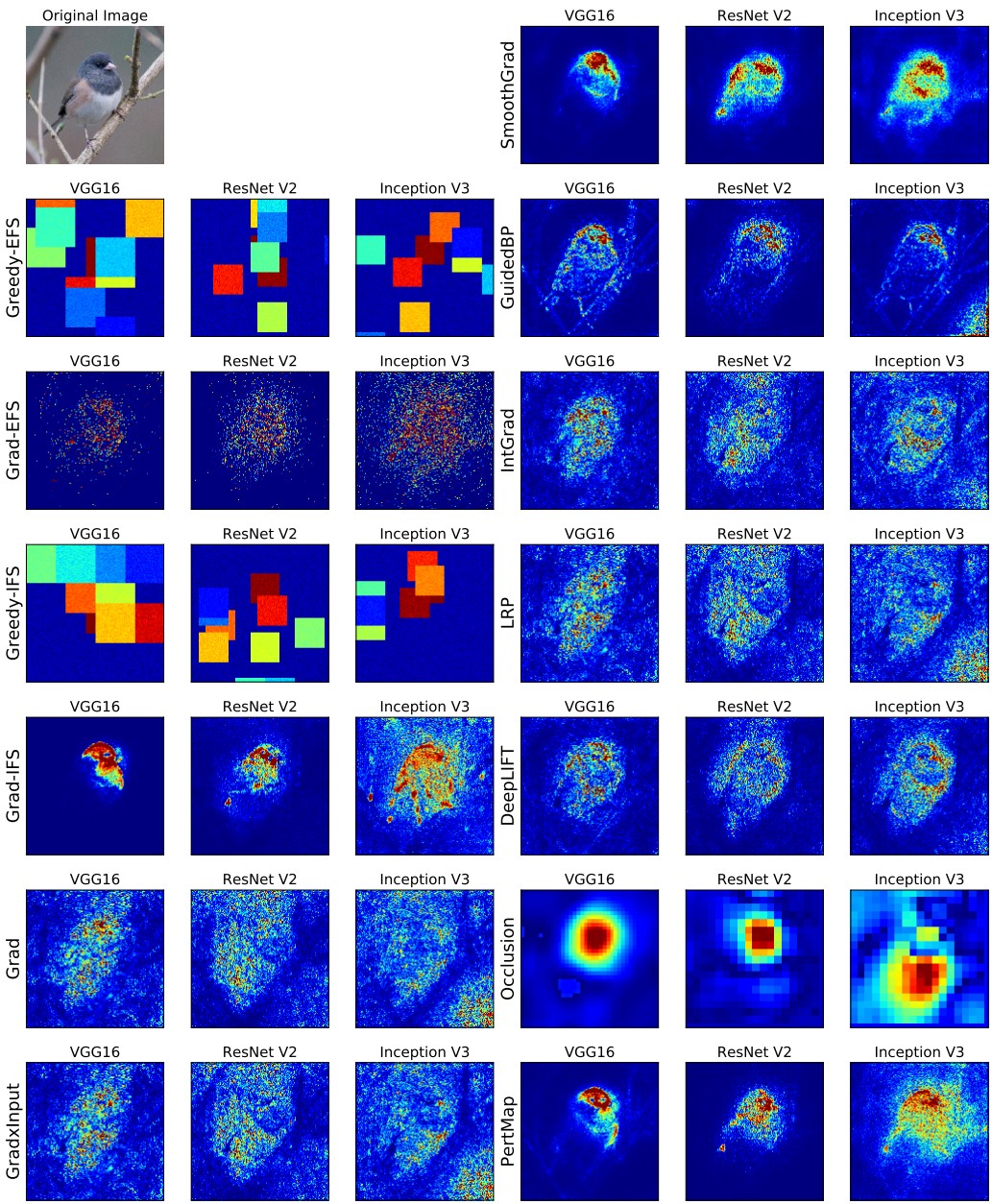

Figure 12: Example Heatmaps: The red colored pixels are found to be strongly relevant with each method.

