# OpenReview forum: "Feature Attribution As Feature Selection"
_ICLR.cc/2019/Conference_

### Official Review · AnonReviewer2 · 2018-11-03
**Limited novelty and technical contribution**

**Rating:** 3
**Confidence:** 3

**Review:**

This paper formulates feature attribution from a feature selection perspective, and compares EFS (Exclusive Feature Selection) and IFS (Exclusive Feature Selection), which shows IFS is a better fit for feature attribution.

[+] The paper is well-structured and the proposed approach is clearly presented.
[-] It would helpful if the author could discuss the time complexity of proposed methods and compare the running time with baseline methods in evaluation.
[-] My major concern on this paper is the significance, as the contribution of the paper seems to be very limited.
    1) Formalizing the feature attribution problem as a feature selection problem is straightforward. IFS and EFS are just Forward and Backward stepwise feature selection, which are classic feature selection schemes. Applying them to feature attribution/saliency map does not seem to have much technical contribution.
    2) One claimed contribution of this paper is that existing feature attribution methods can be viewed as approximation of IFS and EFS. However, this contribution also seems to be minor. As many feature selection methods are known to be approximation of backward or forward stepwise feature selection, it is straightforward to show the connection between other feature attribution methods and IFS/EFS.

In conclusion, I would recommend to reject this paper due to the limited novelty and technical contribution.

---

> ### Author Response · Authors · 2018-11-19
> **Clarifying the connection between feature attribution and feature selection will open up future research directions.**
>
> First of all, we would like to thank you for your time and efforts to review our paper.
>
> > the time complexity
>
> Grad-IFS (which attained the best AOIC) requires solving an optimization problem, and it is therefore not very fast. Practically it takes around a few minutes with one GPU. We are working on improving the computation time, but we leave it beyond the scope of this manuscript because our focus is on *formalizing the feature attribution problem* rather than claiming that our method is SOTA.
> We would also like to mention that *fast computation* is not always the first priority. Feature attribution is used, e.g. for checking whether the model made decisions based on right reasons. For that purpose, less accurate methods are not preferable even though they are computationally fast. Grad-IFS (or PertMap) would be more appropriate for such purpose where they can highlight "the reason" more accurately (as we have demonstrated in the experiments).
>
> > 1) Formalizing the feature attribution problem as a feature selection problem is straightforward. IFS and EFS are just Forward and Backward stepwise feature selection, which are classic feature selection schemes. Applying them to feature attribution/saliency map does not seem to have much technical contribution.
>
> Please remind the fact that while tens of feature attribution methods have been proposed in the last few years, the connection between feature attribution and feature selection was overlooked. Instead, most studies have focused on improving the gradient-based methods (as we can see in Fig1) although, to date, the connection between the gradient and the "relevance" remains unclear. We believe that clarifying the connection between feature attribution and feature selection can enrich the research direction beyond the gradient. This is the reason why we raised the research question "(Q1) how can we define relevance?" Our observation is that there is no reason to stick to the gradient. We believe that (even though it may sound straightforward), our observation is important to push the entire research field forward beyond the gradient.
>
> > 2) One claimed contribution of this paper is that existing feature attribution methods can be viewed as approximation of IFS and EFS. However, this contribution also seems to be minor. As many feature selection methods are known to be approximation of backward or forward stepwise feature selection, it is straightforward to show the connection between other feature attribution methods and IFS/EFS.
>
> We believe that understanding existing feature attribution methods from the feature selection perspective is very useful. For example, in Appendix B, we found that SmoothGrad can be interpreted as one-step GD approximation starting from a non-zero parameter, while other gradient-based methods are one-step GD approximation starting from zeros. This can explain the practical advantage of SmoothGrad, which is known to perform well in practice (and also in AOIC as we have demonstrated). The connection can also open up future research directions for improving feature attribution methods. For example, we can naturally extend the existing gradient-based methods from one-step GD approximation to few-steps GD approximation.

---

### Official Review · AnonReviewer1 · 2018-11-04
**good review but could be lacking in terms of contribution**

**Rating:** 4
**Confidence:** 4

**Review:**

In this paper, the authors study the feature attribution problem as feature selection (exclusive and inclusive).  The authors go through previous work, provide definitions and attempt to answer questions that are relevant to this task.  The authors provide several experiments in order to empirically evaluate which of the two feature selection approaches is better suited for feature attibution.  Although this is a good review, and the motivation is sound, I think that much more ellaboration, and experiments on more than 200 images, would be required to reach definitive conclusions.

---

> ### Author Response · Authors · 2018-11-19
> **We are happy to elaborate the discussion. What should we elaborate?**
>
> First of all, we would like to thank you for your time and efforts to review our paper.
>
> > much more ellaboration
>
> We are happy to elaborate the discussion so that our manuscript to be more useful to the readers. We are happy if you can point out which part of our manuscript needs elaboration. For example, which of our discussions are less convincing or need to be detailed to improve the readability.
>
> > experiments on more than 200 images
>
> To tell the truth, this was because of our budget and time limitation. Even for 200 images, just evaluating AUEC and AOIC takes around two weeks (see the details below). To increase the number of images to be evaluated, we need expensive environments which we cannot manage. Please takes this fact into consideration.
> Here, we would also like to mention that, even with 200 images, Fig4 would be already sufficient to conclude that IFS is advantageous than EFS. We did not select results to bias our conclusion. For 200 images, Grad-EFS produced shot noises (i.e. adversarial examples), while Grad-EFS produced reasonable heatmaps. Moreover, while AUEC was not good at evaluating the performance differences between several methods, AOIC could distinguish good methods and bad methods. These results indicate that IFS is better suited as the formalization of the feature attribution problem.
>
> [Why two weeks?]
> To evaluate one g_c(S_q) in Sec4 for a threshold q, we generated 100 noises as r, and computed the empirical average. This requires 100 forward propagation in DNN. To evaluate AUEC and AOIC, we varied the threshold q for around 30 different values. Thus, for one heatmap, 6000 forward propagation is required to compute both AUEC and AOIC. Because we computed AUEC and AOIC for 200 images with 14 different feature attribution methods, the number of forward propagation is then 16800000 for one DNN. Because we evaluated for three DNNs and for two types of noises r, the total number of forward propagation is 1000800000. Even if one forward pass takes 0.01sec, the total runtime required is 10008000sec ~ 280hours, which is almost two weeks.
> We also note that, this 280 hours is just for evaluating AUEC and AOIC. Our experiments also require certain amount of times for computing heatmaps (thus, the entire experiments can take more than two weeks).

---

### Official Review · AnonReviewer3 · 2018-11-05
**FEATURE ATTRIBUTION AS FEATURE SELECTION**

**Rating:** 4
**Confidence:** 2

**Review:**

The authors formalize the feature attribution problem as a feature selection problem and they demonstrate that several existing feature attribution methods can be interpreted as approximation algorithms for Exclusive Feature Selection and Inclusive Feature Selection.

- The authors claim that IFS is better suited as the formalization for the feature attribution problem and EFS has several unfavourable properties. Although they did exhaustive experiments to show this, it is not clear to the reviewer.

- Also, it is more interesting if the authors can show how we use IFS in real applications.

---

> ### Author Response · Authors · 2018-11-19
> **IFS is better because EFS is quite similar to AE.**
>
> First of all, we would like to thank you for your time and efforts to review our paper.
>
> > - The authors claim that IFS is better suited as the formalization for the feature attribution problem and EFS has several unfavorable properties. Although they did exhaustive experiments to show this, it is not clear to the reviewer.
>
> The reason why EFS is not favorable is because "In EFS (2.2), instead of the data perturbation, one searches for a small number of corrupted features that reduces the class intensity", which is very similar to adversarial example (AE) where one seeks the minimum data perturbation that changes the model’s output. The experimental results also support this observation. This is the reason why we concluded EFS is not favorable.
> We are happy to have more feedbacks where you find it unclear. We will elaborate the discussion in the manuscript based on the feedback.
>
> > it is more interesting if the authors can show how we use IFS in real applications.
>
> All the feature attribution methods mentioned in this paper can be used for the same purposes. For example, as we have done in the experiments, we can used them to highlight where DNN has focused on when making decisions.
> We note that our focus is  on formalizing the feature attribution problem. That is, while several feature attribution methods have been proposed, to date, the formal definition of "relevance" underlying those methods remains unclear. We believe that our study is a first step towards understanding the relevance underlying several feature attribution methods. We also believe that formalizing the relevance is also helpful developing better methods, as we have demonstrated in Grad-IFS which attained the best AOIC.

---

### Meta-Review · Area_Chair1 · 2018-12-15

**Confidence:** 4
**Recommendation:** Reject

**Metareview:**

All in all, while the reviewers found that the problem at hand is interesting to study, the submission's contributions in terms of significance/novelty did not rise to the standards for acceptance. The reasoning is most succinctly discussed by R3 who argues that IFS and EFS are basically feature selection and applying them to feature attribution is not particularly novel from a methodological point of view.